# Identification of Biomarkers Co-Associated with Lactylation and Acetylation in Systemic Lupus Erythematosus

**DOI:** 10.3390/biomedicines13061274

**Published:** 2025-05-22

**Authors:** Zhanyan Gao, Yang Feng, Chenghui Zheng, Fei Li, Zhan Sun, Mengmeng Xiang, Junrong Zhu, Mingyu Chu, Jinhua Xu, Jun Liang

**Affiliations:** 1Department of Dermatology, Huashan Hospital, Fudan University, Shanghai 200040, China; 21211220028@m.fudan.edu.cn (Z.G.); fengyangmd@126.com (Y.F.); 23111220130@m.fudan.edu.cn (C.Z.); b0757@huashan.org.cn (F.L.); 22211220033@m.fudan.edu.cn (Z.S.); 21111220049@m.fudan.edu.cn (M.X.); youyou_wangwang@163.com (J.Z.); 17301050267@fudan.edu.cn (M.C.); 2Department of Dermatology, Shanghai Institute of Dermatology, Huashan Hospital, Fudan University, Shanghai 200040, China

**Keywords:** lactylation, acetylation, systemic lupus erythematosus, CDCA5, MCTS1

## Abstract

**Background:** Systemic lupus erythematosus (SLE) is an immune-mediated disease with widespread involvement, and its pathogenesis remains incompletely understood. Recent studies suggest that modifications such as acetylation and lactylation play crucial roles in SLE progression, with potential interrelationships between them. This study aimed to identify biomarker genes co-associated with both lactylation and acetylation and to explore their potential mechanisms in SLE pathogenesis. **Methods:** Microarray data from peripheral blood mononuclear cells (PBMCs) of SLE patients and healthy controls were obtained from the Gene Expression Omnibus (GEO) database. In the training dataset (GSE81622), differential expression analysis was performed to compare SLE samples with healthy controls. Lactate- and acetylation-related genes were used to identify differentially expressed lactate-related genes (LR-DEGs) and acetylation-related genes (AR-DEGs). Genes co-associated with both lactylation and acetylation were further examined. LASSO regression, support vector machine recursive feature elimination (SVM-RFE), and ROC curve analysis were used to identify hub genes. Immune infiltration analysis and a clinical nomogram model were developed for accurate diagnosis and treatment prediction. qPCR was used to validate the hub genes. **Results:** A total of 1181 differentially expressed genes (DEGs) were identified between SLE and healthy groups. Of these, 33 LR-DEGs and 28 AR-DEGs were identified. Seven genes were found to be co-associated with both lactylation and acetylation. Using LASSO and SVM-RFE, two hub genes, CDCA5 and MCTS1, were identified and validated in the GSE24706 dataset. ROC curve analysis and clinical nomogram revealed significant associations of these biomarkers with SLE pathogenesis. **Conclusions:** Our study identifies CDCA5 and MCTS1 as potential biomarkers for SLE, potentially influencing its pathogenesis through histone lactylation and acetylation. Experimental validation confirmed their differential expression between SLE patients and healthy controls. These findings underscore the role of epigenetic modifications in SLE, offering new insights into its regulatory mechanisms and immune interactions.

## 1. Introduction

Systemic lupus erythematosus (SLE) is a chronic autoimmune disease characterized by aberrant immune responses, autoantibody production, and widespread tissue damage. The precise etiology of SLE remains incompletely understood, but genetic, epigenetic, and environmental factors have been implicated in its pathogenesis. SLE arises from a vicious cycle of innate–adaptive immune dysregulation. Defective clearance of apoptotic debris activates plasmacytoid dendritic cells via TLR7/9, driving pathological type I interferon (IFN) overproduction that primes autoreactive B cells to produce anti-dsDNA antibodies. Concurrently, basophils amplify inflammation through IL-4/IL-6-mediated germinal center support and dendritic cell activation [1,2]. Sustained IFN signatures and B/T cell tolerance breakdown underpin clinical heterogeneity [3]. This may explain why the clinical manifestations of SLE patients are highly heterogeneous, with diverse features of systemic involvement, varying in course and severity of the disease [3,4]. Different patients have different clinical manifestations, and the same patient exhibits different clinical features at different stages of the disease course. The prognosis of patients with the same clinical phenotype also varies. Therefore, the diagnosis and treatment of SLE have always been a challenge.

Among these, post-translational modifications (PTMs) of histones have gained increasing attention due to their crucial role in regulating chromatin dynamics and gene expression. Dysregulation of histone modifications has been linked to immune dysfunction and autoimmunity, suggesting their potential involvement in SLE pathophysiology.

As a terminal product of anaerobic glycolysis, lactate serves not merely as a metabolic byproduct but as a dynamic signaling molecule. Its pleiotropic roles in immunomodulation and epigenetic regulation have garnered significant interest, particularly following the elucidation of Warburg effect-driven metabolic adaptations in malignancies. Protein lactylation is a post-translational modification of proteins first reported in 2019, involving the binding of lactate molecules to lysine residues [5]. Subsequent research further confirms that protein lactylation is an important way for lactate to function, participating in important life activities such as macrophage polarization [6], tumor proliferation [7], nervous system regulation [8], and metabolic regulation [9]. Recent studies have shown that protein lactylation plays a crucial role in systemic lupus erythematosus [10].

Histone acetylation, one of the most extensively studied PTMs, is catalyzed by histone acetyltransferases (HATs) and reversed by histone deacetylases (HDACs) [11]. It plays a pivotal role in chromatin accessibility and transcriptional activation. Previous studies have demonstrated that aberrant histone acetylation contributes to immune cell activation and the upregulation of pro-inflammatory genes in SLE [12]. For instance, increased histone acetylation at the promoters of key immune-related genes was associated with excessive cytokine production and autoreactive T and B cell responses, thereby exacerbating disease severity [13]. Additionally, HDAC inhibitors were shown to modulate immune responses and ameliorate lupus-like symptoms in experimental models, highlighting the therapeutic potential of targeting histone acetylation in SLE [14,15].

The relationship between lactylation and acetylation is inseparable. For example, previous studies mentioned histone hyperacetylation and inactivation enhancing lactate overproduction [16]. Lactate promotes macrophage histone lactylation and acetylation [17], and lysine acetylase p300 was reported to catalyze the transfer of the lactyl group from lactyl CoA to histones in a cell free system [17]. The dysregulation of gene expression associated with both lactylation and acetylation may be closely related to the development and pathogenesis of SLE. And its exact role is still unclear. Therefore, this article aims to explore the relationship between lactylation, acetylation, and SLE, where the identification of biomarkers co-associated with lactylation and acetylation would provide a new insight into SLE diagnosis and treatment.

## 2. Materials and Methods

### 2.1. Study Design and Data Processing

This study employed a multiphase integrative approach combining retrospective bioinformatics discovery with prospective experimental validation to identify lactylation/acetylation-associated hub genes in SLE pathogenesis. The workflow is illustrated in the flow diagram (Figure 1). The GSE81622 [18] and GSE24706 [19] microarray datasets containing gene expression profiles and clinical characteristics data from SLE patients and healthy individuals were downloaded from the GEO database (https://www.ncbi.nlm.nih.gov/geo/, accessed on 6 August 2024). A sum of 419 lactate-related genes (LRGs) and 430 acetylate-related genes (ARGs) were obtained from the Molecular Signatures Database (http://www.gsea-msigdb.org/gsea/index.jsp, accessed on 6 August 2024).

### 2.2. Indentification of DEGs and Enrichment Analysis

In this study, the limma package in R [20] was utilized to extract the differentially expressed genes (DEGs) between SLE patients and healthy individuals. The threshold was set as |log_2_ fold change (FC)| > 1 and adjusted *p* value < 0.05. Volcano plots were applied to show DEGs via the ggplot2 package in R. Following this, Gene Ontology (GO) analysis, Kyoto Encyclopedia of Genes and Genomes (KEGG) analysis, and Disease Ontology (DO) analysis were intended to conduct in-depth research on the function characteristics and molecular pathways related to DEGs. The clusterProfiler package in R was used for functional enrichment analysis and exploring the possible mechanisms by which DEGs participate in SLE [21]. Additionally, the co-expression network of DEGs was constructed using GeneMANIA (http://www.genemania.org/, accessed on 18 March 2025).

### 2.3. The Acquisition of Genes Which Are Co-Associated with Lactylation and Acetylation

The genes common to both DEGs and LRGs were classified as lactate-related differential expression genes (LR-DEGs), as shown in the Venn diagram. Similarly, the genes at the intersection of DEGs and ARGs were defined as acetylate-related differential expression genes (AR-DEGs). Further, we acquire the genes which are co-associated with lactylation and acetylation. The chromosome localization of these co-associated genes was visualized using Rcircos (1.2.2) [22]. A protein–protein interaction (PPI) network was constructed using the STRING database (confidence > 0.7) to investigate interactions among these genes. GO and KEGG enrichment analysis of these genes was conducted via the clusterProfiler package in R [21], where the *p*.adjust.value < 0.05 was set as the significance threshold.

### 2.4. Machine Learning Algorithms

The co-association genes between SLE patients and healthy individuals are considered a variable in two machine learning algorithms: the least absolute shrinkage and selection operator (LASSO) and support vector machine recursive feature elimination (SVM-RFE) for filtering important diagnostic variables in the training set (GSE81622) [23,24]. The key genes were obtained by pooling the results of these two algorithms. LASSO regression was implemented using the glmnet package in R, with a 10-fold cross validation to determine the optimal lambda value, which controls feature selection and model sparsity. SVM-RFE was employed to iteratively remove the least important features based on weight rankings, improving classification performance.

To evaluate the diagnostic efficiency of the identified genes, we performed receiver operating characteristic (ROC) curve analysis using the pROC package in R [25]. The area under the ROC curve (AUC) was calculated to assess each gene’s diagnostic potential. Genes with AUC > 0.7 were considered high-value diagnostic markers and were subsequently validated in an independent dataset (GSE24706). If these genes demonstrated consistent expression trends in both datasets, they were recognized as potential biomarkers for SLE.

Prior to statistical analysis, we assessed the normality of gene expression data using the Shapiro–Wilk test. If the data were normally distributed, parametric tests (e.g., *t*-test, Pearson correlation for covariation analysis) were applied; otherwise, nonparametric tests (e.g., Mann–Whitney U test, Spearman correlation) were used.

### 2.5. Clinical Nomogram Model

The nomogram takes complex regression equations and transforms them into visual graphs that make the results of predictive models more readable and easy to assess for patients and clinicians [26]. It has been widely used to predict the probability of the individual occurrence of clinical events. The nomogram was generated using the rms package in R, integrating key diagnostic genes and other relevant clinical features to estimate SLE probability. The performance of the nomogram was assessed through calibration curves, comparing predicted vs. observed probabilities. ROC curve analysis was performed to evaluate the model’s discriminative ability (AUC). To further verify the stability and reliability of the nomogram, we conducted bootstrapping with 1000 resamples on PBMC-derived transcriptomic data for internal validation. All statistical analyses were conducted using R software (version 4.3.2), and a *p*-value < 0.05 was considered statistically significant.

### 2.6. Immune Cell Infiltration Analysis

Single sample gene set enrichment analysis (ssGSEA) was conducted to estimate the relative abundance of different immune cell types in each sample through the GSVA package in R [27]. The CIBERSORT algorithm was applied to explore the relative abundance of 22 immune cells infiltrated in the SLE microenvironment [28]. Finally, pheatmap was used to visualize the immune cell infiltration in different samples and a single characteristic gene.

### 2.7. Construction of Regulatory Networks

The microRNA (mi-RNA) and transcription factor (TF) regulatory networks linked to these biomarkers were explored and constructed using NetworkAnalyst [29] (http://www.networkanalyst.ca, accessed on 18 March 2025).

### 2.8. Human Blood Specimen Collection

Peripheral blood samples were collected from individuals diagnosed with SLE (*n* = 14), with control samples being collected from healthy individuals (*n* = 10), between January 2024 and March 2025. These participants were enlisted from the out-patient clinic of the Department of Dermatology, Huashan Hospital, Fudan University. All patients included in the study strictly followed the 2019 EULAR/ACR classification criteria [30] and excluded pregnant and lactating women; patients who participated in other clinical trials or used any biological agents for the treatment of SLE within 3 months and/or currently; or patients who have other major primary diseases. Compliance with the Declaration of Helsinki was ensured for all procedures involving human participants. Ethical approval was obtained from the Institutional Review Board of Huashan Hospital, Fudan University, on 26 October 2021 (Approval No. 2021-879). All participants provided written informed consent prior to their inclusion in this study. The demographic and clinical characteristics of SLE patients and healthy individuals are displayed in Appendix A.

### 2.9. Quantitative PCR Assay

Total RNA was extracted using RNAiso Plus reagent (Takara, Kusatsu, Japan), followed by genomic DNA elimination and reverse transcription with a PrimeScript™ RT Reagent Kit (Takara, Kusatsu, Japan). Quantitative PCR amplification was performed on a Bio-Rad real-time PCR apparatus (California, CA, USA) with TB Green Premix Ex Taq II (Takara, Kusatsu, Japan), using GAPDH primers (Sangon Biotech, Shanghai, China) as an internal control. Relative mRNA expression levels were calculated through relative quantification, with primer sequences provided in Appendix A.

### 2.10. Statistical Analysis

All demographic and clinical data of SLE patients and healthy individuals were assessed for normality of data distribution using the Shapiro–Wilk test (*p* > 0.05 threshold), and homogeneity of variance was verified using the Levene test. Group comparisons were performed using parametric tests (independent *t* test or Welch *t* test for unequal variances) or the nonparametric Mann–Whitney U test, as appropriate. Categorical variables (e.g., sex, comorbidity rate) were analyzed using Fisher’s exact test. To address the issue of mismatch in age and BMI, we included these variables as covariates in the ANCOVA model. Confounding variables were further screened by stepwise regression (*p* < 0.1 entry threshold), and multicollinearity was excluded (VIF < 2.5). Multiple comparisons were adjusted by Bonferroni correction where applicable. All analyses were performed in SPSS 26.0 (α = 0.05, two-tailed).

Differences in gene expression levels between groups were analyzed using independent samples *t*-tests. All statistical analyses were performed using R version 4.3.2 and GraphPad Prism 9.0 (GraphPad Software, San Diego, CA, USA). Statistical significance was defined as follows: * *p* < 0.05 (single asterisk), ** *p* < 0.01 (double asterisks), *** *p* < 0.001 (triple asterisks), **** *p* < 0.0001 (quadruple asterisks). And *p*-value < 0.05 was considered statistically significant.

## 3. Results

### 3.1. Identifications of DEGs Between SLE Patients and HCs

Utilizing the limma package in R, 1181 DEGs were pinpointed from the dataset, with 233 genes being upregulated and 174 genes being downregulated in SLE patients relative to healthy individuals (Figure 1B). We explored differences in the expression of genes co-associated with 430 ARGs and 419 LRGs between the SLE patients (*n* = 30) and healthy individuals (*n* = 25) in the GSE81622 training dataset. There were seven significantly dysregulated co-associated genes including CDCA5, OXNAD1, H1-2, MCTS1, RIN2, TXN, and NAT10 (Figure 1A). The chromosomal positions of these seven co-associated genes were visualized via the Rcircos package in R (Figure 1C).

### 3.2. Functional Enrichment Analysis of Co-Associated DEGs

Functional enrichment analysis was conducted to further explore the function of these seven co-associated genes in the SLE patients. GO enrichment analysis (including biological process (BP), cellular component (CC), and molecular function (MF)) showed that the co-associated genes were mainly involved in chromosome condensation, DNA binding, packaging, organization, and metabolic processes and were also enriched in iRES-dependent viral translational initiation and establishment of sister chromatoid cohesion (Figure 2B). KEGG analysis showed that these genes were predominantly enriched for ribosome biogenesis in eukaryotes, fluid shear stress, and atherosclerosis (Figure 2D). DO enrichment analysis showed that these DEGs were associated with cutis laxa, blast benign neoplasm and thoracic benign neoplasm, intermediate coronary syndrome, myocarditis, toxic encephalopathy, extrinsic cardiomyopathy, diabetic neuropathy, alopecia, hypotrichosis, rhabdomyosarcoma, hair disease, skeletal muscle cancer, and sensorineural hearing loss (Figure 2E). Construction of co-expressed gene networks using GeneMANIA revealed a complex PPI network (Figure 2F,G). Then, we used the corrplot package in R to explore the correlation between the seven hub genes (Figure 2H). These results elucidated that these DEGs are closely related to various signaling processes.

### 3.3. Identification and Validation of SLE-Associated Lactylation and Acetylation Regulators

First, LASSO regression was performed on the seven identified co-associated DEGs, which led to feature selection, yielding five optimal variables, which were modeled to have an AUC value of 0.968 (Figure 3A–C). SVM-RFE was used to find optimal variables by removing the generated feature vectors, which led to the identification of five related genes as well (Figure 3D–F). To find the most relevant SLE regulators, the marker genes from the two algorithms were intersected, presenting four key genes, CDCA5, OXNAD1, H1-2, and MCTS1, of which CDCA5 was the most important marker (Figure 3G–I).

To further test the diagnostic accuracy of these co-associated genes, ROC analysis was performed and validated on the GSE24706 dataset. CDCA5 and MCTS1 showed the same expression trend in the validation dataset as in the training set, and these genes were significantly higher in SLE samples than in the healthy individual samples based on peripheral blood samples (Figure 4A). ROC analysis also showed that CDCA5 and MCTS1 have significance to distinguish between SLE patients and healthy individuals with relatively high AUC values of 0.867 and 0.827, respectively. High diagnostic values were also achieved in the GSE24706 validation dataset with an AUC of 0.773 and 0.830, respectively (Figure 4B).

Then, NetworkAnalyst was used to further investigate those miRNAs and TFs that might be involved in regulating the expression of these genes. The results showed that MCTS1-associated miRNAs were the most abundant, while the TF target network showed that CDCA5 produced the most abundant TF diagnostic biomarker pairs (Figure 4C–E).

### 3.4. Construction of Clinical Nomogram

The clinical nomogram model was conducted by fitting two candidate genes, CDCA5 and MCTS1, and projecting each variable in the nomogram plot upward to a point (Figure 5A). The sum of the points of the two variables was converted into an individual disease risk, where the higher the total score, the higher the risk of SLE. Calibration curves showed no significant deviation between actual and predicted observations (Figure 5B). The clinical utility of the nomogram was validated using DCA and CIC. DCA showed better overall net benefit with a threshold probability of 0–1, and CIC also performed well over the entire range of threshold probabilities (Figure 5D).

### 3.5. Evaluation of Immune Cell Infiltration

Immune and inflammatory responses are crucial in the molecular pathways of SLE. To further explore the relationship between the lactylation, acetylation, and the immune microenvironment of SLE, we used ssGSEA to calculate the abundance of immune cells in PBMCs from SLE patients and evaluate the specific enrichment level of two hub genes.

Compared with healthy individuals, we found significantly higher infiltration with monocytes, regulatory T cells (Tregs), naïve B cells, and plasma cells in SLE patients than in healthy groups, whereas CD4 memory resting T cells and resting NK cells were more abundant in healthy individuals (Figure 6A,B and Figure 7A). Notably, Figure 6A presents the global immune landscape using unsupervised hierarchical clustering of all samples, while Figure 7A focuses on group-wise statistical comparisons between SLE and healthy controls. The difference in visualization strategies accounts for the apparent variation in presentation.

Finally, we analyzed the correlation between the two previously identified hub genes and immune cell types. We found that MCTS1 and CDCA5 were significantly positively correlated with plasma cells, Tregs, and monocytes (Figure 7B).

### 3.6. Quantitative PCR Results

The findings of qRT-PCR were in line with the genomic analysis. In the comparison between SLE and healthy groups, the expression level of CDCA5 and MCTS1 were significantly upregulated in SLE (Figure 7C,D). Combining the previous results, it suggests that CDCA5 and MCTS1 may be important diagnostic markers for SLE. Although OXNAD1 and H1-2 were not selected as final hub genes, we additionally validated their expression using qRT-PCR. The results showed that OXNAD1 expression was significantly upregulated in SLE patients compared to controls (*p* < 0.001), consistent with transcriptomic findings. However, H1-2 did not exhibit a statistically significant difference between the two groups. These results are shown in Appendix A.

## 4. Discussion

In this study, we identified CDCA5 and MCTS1 as potential biomarkers associated with both histone lactylation and acetylation in SLE and demonstrated their significant upregulation in PBMCs from SLE patients. The expression levels of these genes were validated across independent datasets and further confirmed by qPCR, suggesting their robust association with SLE pathophysiology. Through comprehensive immune infiltration analysis, we observed that both genes were correlated with specific immune cell populations, highlighting their possible involvement in immune dysregulation. These findings offer novel insights into the epigenetic and immunological mechanisms underlying SLE.

Lactylation and acetylation are two metabolically linked post-translational modifications (PTMs) increasingly recognized for their roles in modulating immune responses. Prior studies demonstrated that high acetylation and inactivation of PDHA1 may contribute to excessive lactate accumulation in renal tubular epithelial cells [16,31], a feature relevant to the metabolic reprogramming seen in autoimmune inflammation. Histone acetylation was shown to regulate T cell activation and inflammatory gene expression, and its dysregulation is associated with disease activity in SLE [12,32]. Conversely, histone lactylation—although less studied—was implicated in macrophage polarization and suppression of Treg differentiation [33,34]. Our findings build on this growing literature by identifying CDCA5 and MCTS1 as genes associated with both lactylation and acetylation, suggesting potential crosstalk between these two PTMs in SLE pathogenesis. Notably, CDCA5 may promote H3K18 lactylation at regulatory regions of genes such as Foxp3, potentially impairing immune tolerance by inhibiting Treg differentiation [34,35].

Moreover, CDCA5 may co-activate LDHA, a key glycolytic enzyme, thereby increasing lactate production and reinforcing a pro-inflammatory microenvironment. Elevated lactate not only fuels lactylation but also inhibits SIRT1, a NAD^+^-dependent deacetylase, disrupting the acetylation balance of histone and nonhistone proteins [36]. This disruption could contribute to mitochondrial dysfunction and promote the differentiation of pathogenic Th17 cells via stabilization of HIF-1α. Although these mechanistic links remain speculative, they provide a plausible framework to interpret our observed gene expression patterns and warrant further validation through targeted functional assays such as ChIP-seq or CRISPR-mediated gene editing. Taken together, our results suggest that the interplay between lactylation and acetylation may shape the immune landscape in SLE, positioning CDCA5 and MCTS1 as potential mediators of epigenetic immune dysregulation.

The consistent upregulation of CDCA5 and MCTS1 across multiple datasets, along with their strong diagnostic performance as indicated by ROC and nomogram analyses, underscores their potential as reliable biomarker candidates in SLE. These genes may reflect a broader dysregulation of metabolic–epigenetic circuits within immune cells. However, the specificity of their expression patterns to SLE remains uncertain. It is possible that these changes reflect a shared signature of autoimmune activation rather than a disease-specific hallmark. To clarify this, comparative transcriptomic analyses involving other rheumatic diseases—such as rheumatoid arthritis (RA), Sjögren’s syndrome (SS), and systemic sclerosis (SSc)—are needed [37]. Such investigations would help determine whether CDCA5 and MCTS1 are uniquely associated with SLE or represent common elements of immune dysregulation across autoimmune conditions, thereby improving the precision and clinical applicability of these biomarkers.

Our immune infiltration analysis revealed increased abundance of monocytes, regulatory T cells, naïve B cells, and plasma cells in SLE. However, we did not observe a significant elevation in neutrophils, which appears discordant with previous findings on the expansion of low-density granulocytes (LDGs) in SLE [38]. This discrepancy may be attributed to methodological limitations, as LDGs are functionally distinct and may not be captured accurately by algorithms like CIBERSORT using bulk RNA-seq data. Similarly, basophils—another cell type implicated in SLE pathogenesis—were not detected at significant levels in our analysis, likely due to their low abundance and limited resolution in bulk transcriptomic data. Basophils have been shown to infiltrate peripheral tissues in SLE patients, including the kidneys [39], skin lesions [40], and lymphoid organs [41]. Moreover, multiple studies reported reduced circulating basophil counts in both pediatric [42] and adult SLE patients [43], particularly during active disease phases. These findings highlight the dual dynamics of tissue recruitment and peripheral depletion of basophils in SLE. The apparent absence of basophil signals in our CIBERSORT-based analysis may thus reflect both technical challenges and biological phenomena—namely, the sequestration of basophils in inflamed tissues rather than their presence in the peripheral blood, which formed the basis of our RNA-seq datasets.

Despite the novelty of our findings, there are several limitations to consider. First, there was a notable age disparity between SLE patients and healthy controls (29 ± 6 vs. 39 ± 17 years). Although statistical adjustments were applied to mitigate potential confounding, residual age-related effects cannot be entirely excluded. Prior studies have shown that age influences immune cell composition, inflammatory status, and epigenetic modifications, all of which may affect disease pathogenesis and biomarker expression in SLE [44,45]. Future studies using age-matched cohorts will be necessary to validate the diagnostic utility of CDCA5 and MCTS1.

Second, the current analysis lacks detailed clinical covariates, such as disease duration, medication history, and comorbidities, which may influence the diagnostic value of the identified biomarkers. Once these clinical covariates become available, further statistical modeling including logistic regression or Cox proportional hazards analysis will be performed to determine whether CDCA5 and MCTS1 serve as independent biomarkers for SLE.

Furthermore, while our nomogram demonstrated an AUROC of 0.864 in the discovery dataset, it is important to acknowledge that AUROC values in discovery datasets often tend to overestimate real-world performance due to potential overfitting and dataset-specific biases. Although the three male patients included in our cohort lacked clinical features typically associated with monogenic lupus—such as early-onset disease or a strong family history—this assumption remains speculative without molecular confirmation. Future studies incorporating whole-exome or targeted genetic sequencing would help to accurately classify such cases and determine whether our findings extend to monogenic SLE.

Finally, our immune infiltration analysis was based on bulk RNA-seq data and computational deconvolution, which may have limited resolution for rare immune cell subsets or subtle changes in cellular heterogeneity. Integration with single-cell transcriptomics or spatial profiling could enhance the accuracy of immune landscape characterization in future studies.

## 5. Conclusions

Our study identifies CDCA5 and MCTS1 as potential biomarkers for SLE, likely involved in disease pathogenesis through epigenetic mechanisms such as histone lactylation and acetylation. Their upregulation in peripheral blood and association with immune dysregulation support their relevance to autoimmune processes. These findings highlight the diagnostic promise of epigenetic markers in SLE, though further validation in larger cohorts and mechanistic studies is warranted to confirm their clinical utility.

## Figures and Tables

**Figure 1 biomedicines-13-01274-f001:**
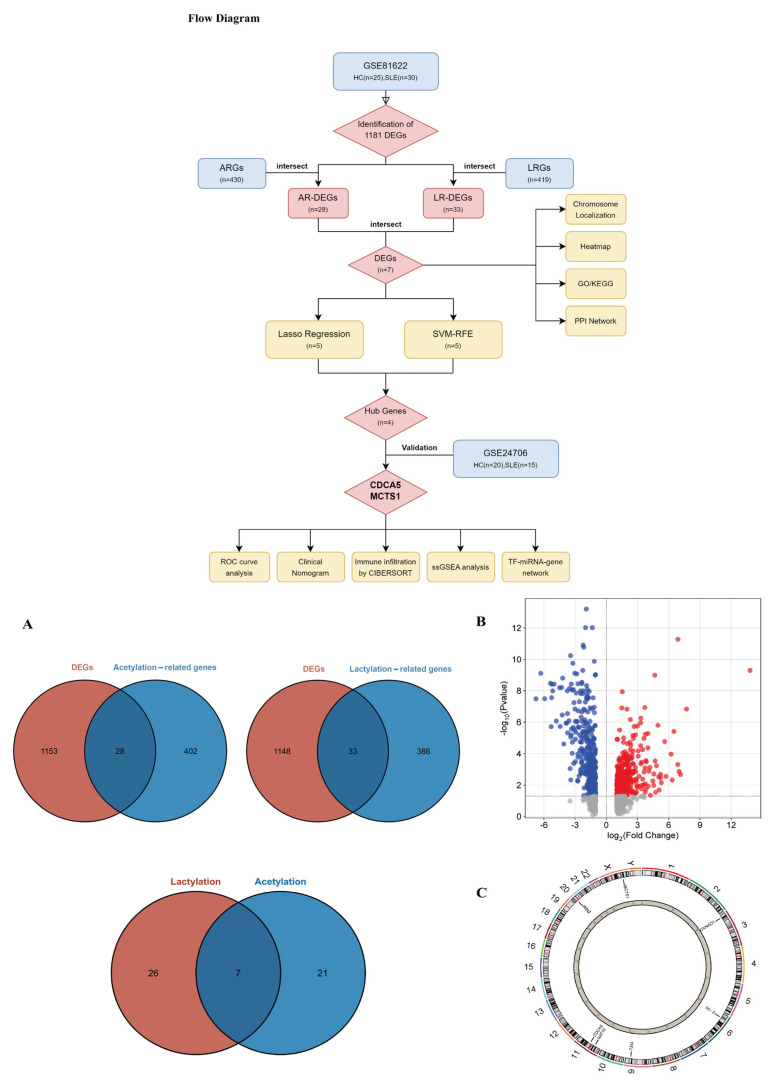
Flow diagram and identification of differentially expressed genes, ARGs, and LRGs. The flow diagram summarizes the content and design flow of this study. (**A**) The Venn diagram shows the DEGs, ARGs, LRGs, and hub genes co-associated with lactylation and acetylation. (**B**) The volcano map shows the genes with |logFC| ≥ 1 and adjusted. *p* value < 0.05 in GSE81622. Red dots represent significantly upregulated genes (log_2_Fold Change > 1 and *p* < 0.05); Blue dots represent significantly downregulated genes (log_2_Fold Change < −1 and *p* < 0.05); Grey dots indicate non-significant genes. (**C**) The chromosomal positions of the seven co-associated genes.

**Figure 2 biomedicines-13-01274-f002:**
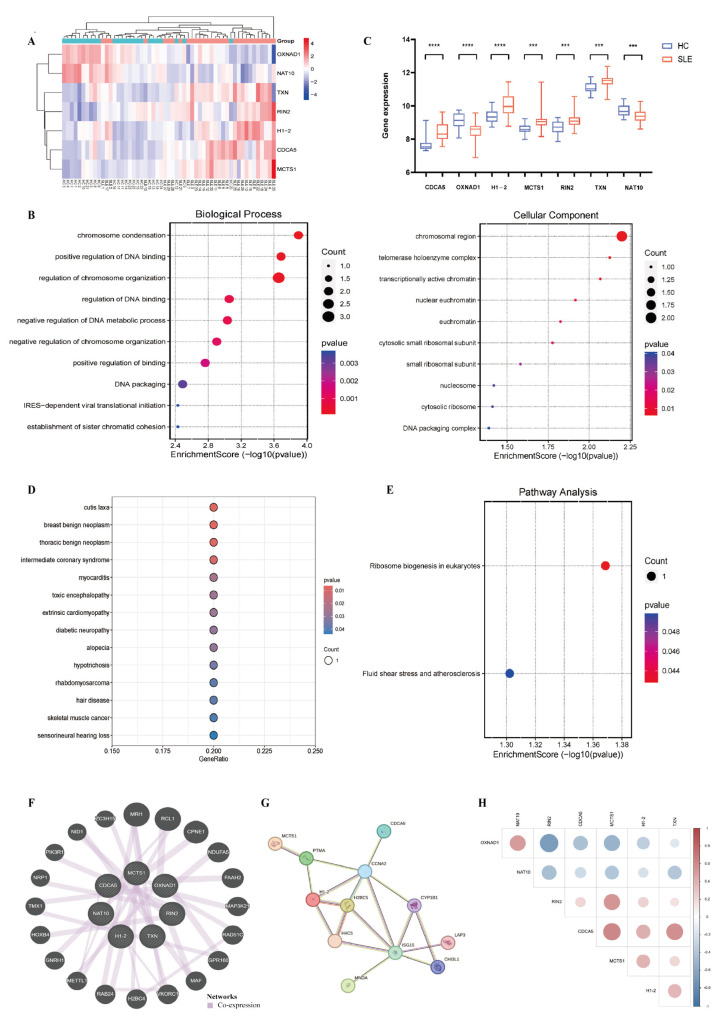
Enrichment analysis to determine the biological characteristics of the seven co-associated genes. (**A**,**C**) The gene expression of seven co-associated genes in PBMC of SLE patients and healthy groups (*** *p* < 0.001, **** *p* < 0.0001). (**B**) The enriched terms in biological process (BP), cellular component (CC), and molecular function (MF) of the genes. (**B**,**D**,**E**) The KEGG and DO enrichment analyses of seven co-associated genes. (**F**) The 7 co-associated genes and their co-expressed genes were analyzed using GeneMANIA. (**G**) The PPI network was constructed to illustrate the interactions among the seven key genes. (**H**) The correlation analysis between the seven co-associated genes.

**Figure 3 biomedicines-13-01274-f003:**
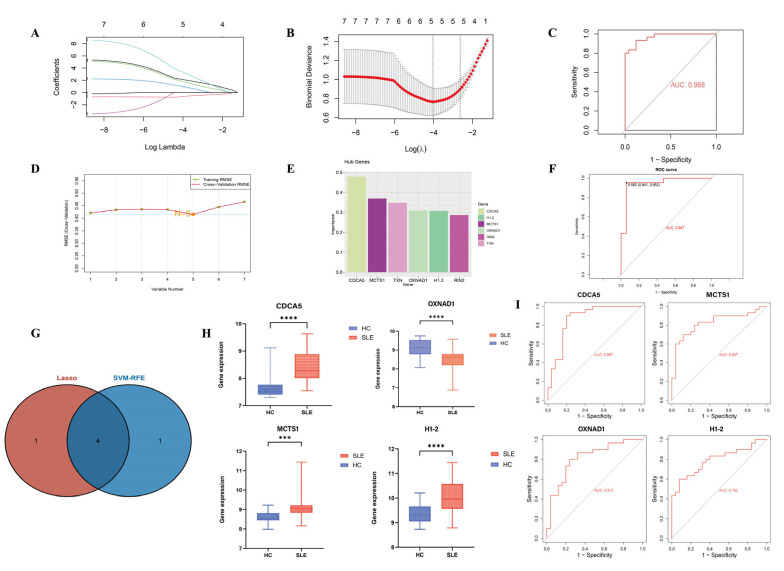
Screening the biomarkers of SLE. (**A**,**B**) The LASSO coefficient profiles and the partial likelihood deviance of co-associated genes in SLE. Five genes were selected at the value (lambda.min). (**C**) The ROC curve analysis of the LASSO regression. (**D**–**F**) SVM-RFE analysis of the co-associated genes and the ROC curve analysis of the model (AUC > 0.7). (**G**) Venn diagram demonstrating the four co-associated genes shared by the LASSO and SVM-RFE algorithms. (**H**) The gene expression level of CDCA5, MCTS1, OXNAD1, and H1-2 in SLE and heathy groups (*** *p* < 0.001, **** *p* < 0.0001). (**I**) The ROC analysis of each gene (CDCA5, MCTS1, OXNAD1, H1-2).

**Figure 4 biomedicines-13-01274-f004:**
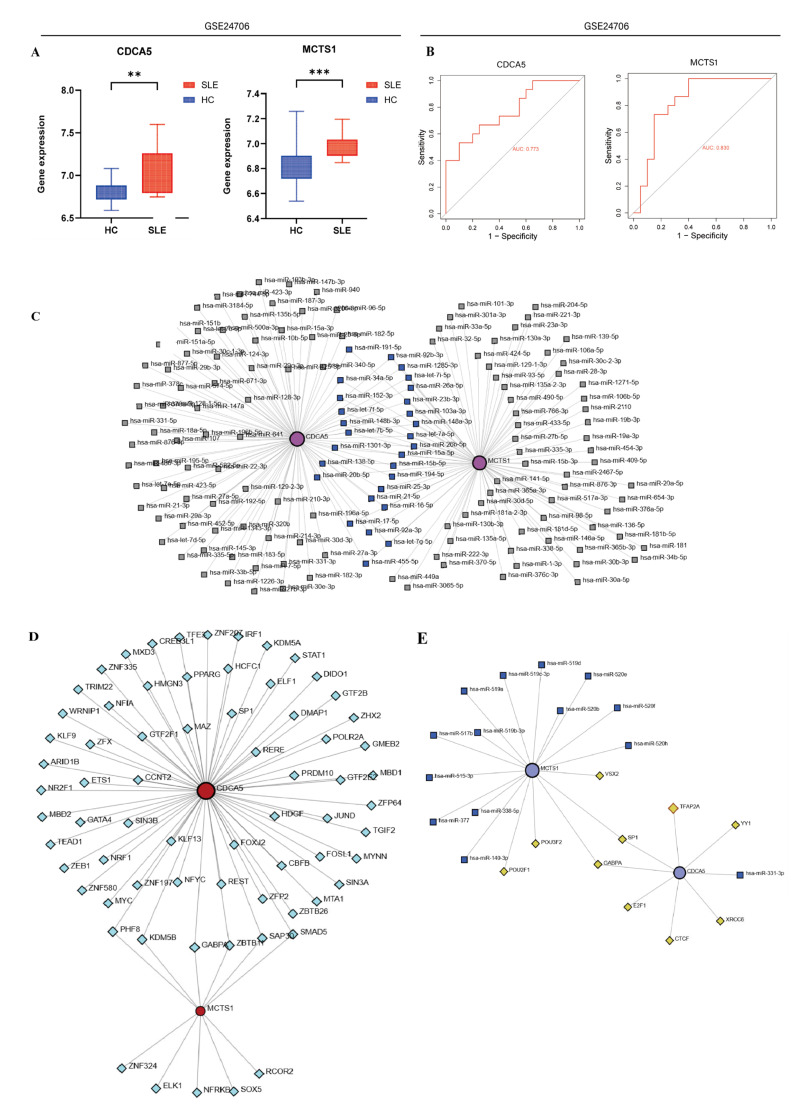
Validation of the diagnostic value of the two-gene signature. (**A**,**B**) The gene expression and ROC curve analysis of CDCA5 and MCTS1 in the validation dataset GSE24706. The ROC curves illustrate the diagnostic performance of CDCA5 and MCTS1. The area under the curve (AUC) for CDCA5 is 0.773, while the AUC for MCTS1 is 0.830 (** *p* < 0.01, *** *p* < 0.001). (**C**–**E**) The miRNA–miRNA network of CDCA5 and MCTS1, illustrating the regulatory interactions between microRNAs (miRNAs) associated with these two genes. The TF–mRNA network of CDCA5 and MCTS1, depicting the transcription factor (TF) regulatory relationships.

**Figure 5 biomedicines-13-01274-f005:**
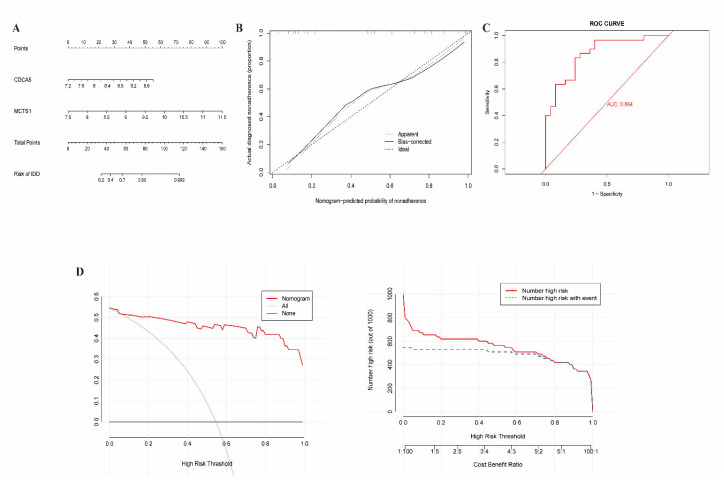
Construction of the nomogram model. (**A**) The ordinary nomogram predicts disease risk of SLE based on CDCA5 and MCTS1. (**B**) Calibration curve for nomogram validation. The calibration curve evaluates the agreement between the predicted probabilities from the nomogram and the actual observed outcomes. The abscissa represents the predicted event occurrence rate, and the ordinate is the observed actual event occurrence rate, with a range of 0–1. (**C**) ROC curve analysis of the nomogram model. The ROC curve assesses the diagnostic performance of the nomogram model. The area under the curve (AUC) reflects the model’s ability to distinguish SLE patients from controls. The AUC of CDCA5 and MCTS1 is 0.864. (**D**) Decision curve analysis (DCA) and clinical impact of the nomogram model. DCA evaluates the clinical utility of the nomogram by quantifying the net benefit at different threshold probabilities.

**Figure 6 biomedicines-13-01274-f006:**
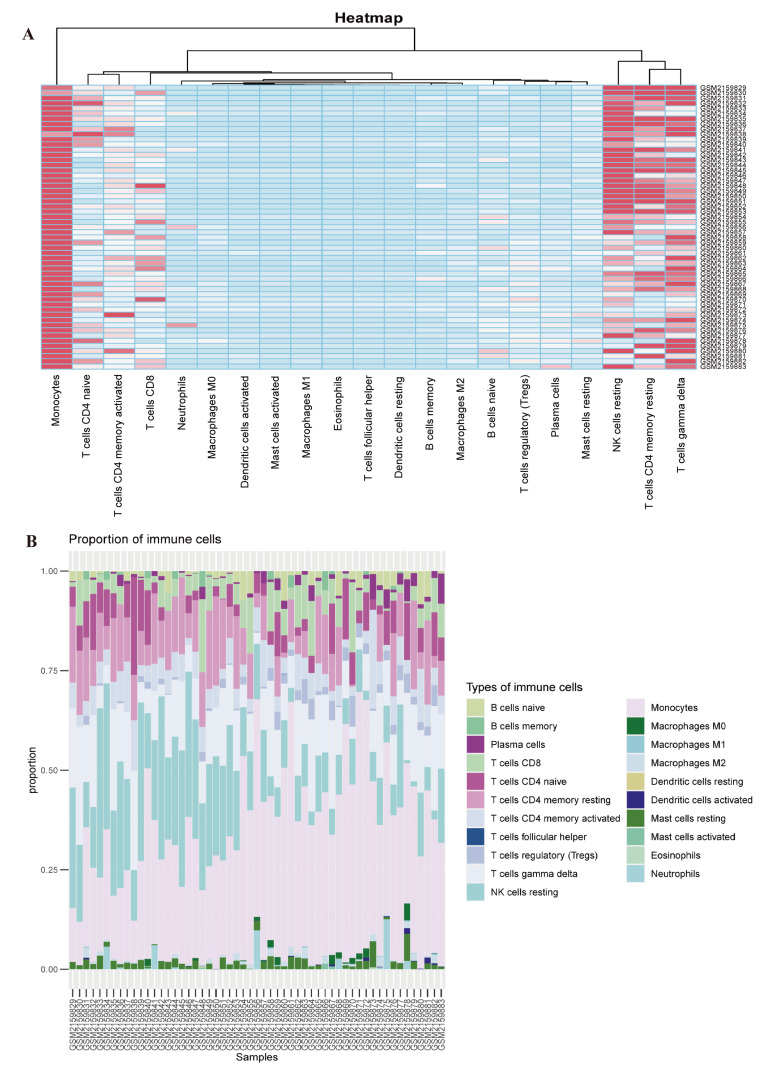
Analysis of immune infiltration in SLE. (**A**,**B**) The distribution of 21 immune cell types in SLE patients and healthy control samples, illustrating differences in immune cell composition between the two groups.

**Figure 7 biomedicines-13-01274-f007:**
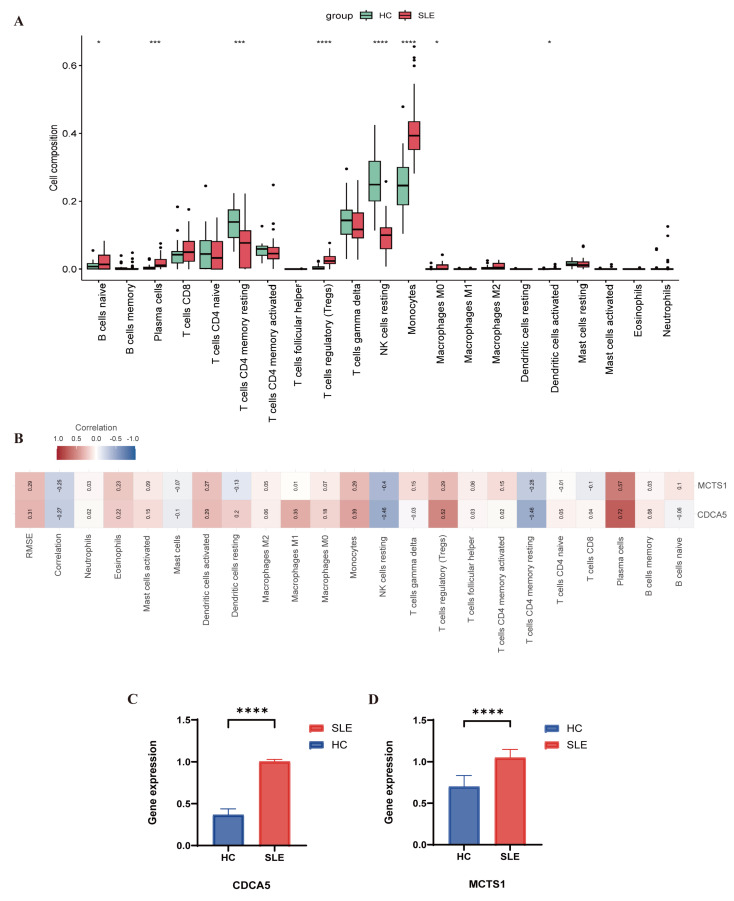
Analysis of immune infiltration and gene expression in SLE. (**A**) The distribution of 21 immune cell types in SLE patients and healthy control samples, illustrating differences in immune cell composition between the two groups. (**B**) Correlation analysis between CDCA5, MCTS1, and infiltrating immune cells in SLE and HC samples. Positive and negative correlations are indicated, with statistical significance denoted by asterisks (* *p* < 0.05, *** *p* < 0.001, **** *p* < 0.0001). (**C**,**D**) qRT-PCR analysis of CDCA5 and MCTS1 mRNA expression levels in PBMCs from SLE patients and healthy individuals. Statistical significance is indicated as follows: * *p* < 0.05, *** *p* < 0.001, **** *p* < 0.0001.

## Data Availability

The datasets presented in this study are available in the GEO database (http://www.ncbi.nlm.nih.gov/geo, accessed on 20 December 2024), GSE81622 and GSE24706.

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
