# Peer review of "Identification of Biomarkers Co-Associated with Lactylation and Acetylation in Systemic Lupus Erythematosus"

_biomedicines, 2025, doi:10.3390/biomedicines13061274_

Round 1
Reviewer 1 Report (Previous Reviewer 2)
Comments and Suggestions for Authors
In general, the authors addressed most of my previous comments.
However, there are still some points that should be better clarified, in my opinion.
For instance, in section 3.5 of the results, the authors state: “Immune and inflammatory responses are crucial in the molecular pathways of SLE. To further explore the relationship between the lactylation, acetylation and the immune microenvironment of SLE, we use ssGSEA to calculate the abundance of immune cells in SLE samples and evaluate the specific enrichment level of two hub genes…”. What exactly are these “SLE samples”? I mean, which tissue do they refer to? I could not find clear answer in the methods either.
Indeed, following the previous point, the issue of the immune infiltration in peripheral (?) tissues is important. The authors state: “Compared with SLE and healthy individuals, we found that there was significantly greater infiltration with monocytes, T cells regulatory, B cells naïve and plasma cells in SLE than in healthy groups. Conversely, there was significantly greater infiltration with T cell CD4 memory resting and NK cell resting in healthy individuals”. Interestingly, I do not see any mention to basophils, which have been described by several groups as an immune cells infiltrating peripheral tissues of SLE patients, for instance, in lupus nephritis (see: PMID: 20825323 ), skin lesions (see: PMID: 28954264) or lymphoid tissues (see: PMID: 20512127). Accordingly, blood basophils are lower in active SLE patients, both children (refer to: PMID: 35885605) and adults (PMID: 25403252). I would like to know how the authors can reconciliate these basophil alteration in the literature with the results of their analysis in terms of immune cells infiltration. Then, the authors should include these aspects in their discussion, also to compare the findings of their analysis with the literature evidence.
The discussion should be revised. This section should start with a list of their main findings, instead of a kind of second introduction. Then, each point should be discussed individually. Moreover, the previous comment should be also used to enrich the discussion with the related supportive references. Finally, a professional editing is important to improve the clarity.
The conclusion is too long and should summarize the main take home message.
Comments on the Quality of English Languagesee comments above
Author Response
Comment1:For instance, in section 3.5 of the results, the authors state: “Immune and inflammatory responses are crucial in the molecular pathways of SLE. To further explore the relationship between the lactylation, acetylation and the immune microenvironment of SLE, we use ssGSEA to calculate the abundance of immune cells in SLE samples and evaluate the specific enrichment level of two hub genes…”. What exactly are these “SLE samples”? I mean, which tissue do they refer to? I could not find clear answer in the methods either.
Response:1:We thank you for pointing out this important clarification. The “SLE samples” referred to in our analysis are peripheral blood mononuclear cell (PBMC) samples from human subjects. We acknowledge that this was not explicitly stated in the original manuscript and could have caused confusion. We have now revised the relevant sections in both the Methods and Results to clearly indicate that all transcriptomic and immune cell infiltration analyses were based on human PBMCs. The text has been updated accordingly to improve clarity.
Comment2: Indeed, following the previous point, the issue of the immune infiltration in peripheral (?) tissues is important. The authors state: “Compared with SLE and healthy individuals, we found that there was significantly greater infiltration with monocytes, T cells regulatory, B cells naïve and plasma cells in SLE than in healthy groups. Conversely, there was significantly greater infiltration with T cell CD4 memory resting and NK cell resting in healthy individuals”. Interestingly, I do not see any mention to basophils, which have been described by several groups as an immune cells infiltrating peripheral tissues of SLE patients, for instance, in lupus nephritis (see: PMID: 20825323 ), skin lesions (see: PMID: 28954264) or lymphoid tissues (see: PMID: 20512127). Accordingly, blood basophils are lower in active SLE patients, both children (refer to: PMID: 35885605) and adults (PMID: 25403252). I would like to know how the authors can reconciliate these basophil alteration in the literature with the results of their analysis in terms of immune cells infiltration. Then, the authors should include these aspects in their discussion, also to compare the findings of their analysis with the literature evidence.
Response 2:We sincerely appreciate your insightful comments regarding the role of basophils in SLE pathogenesis. Indeed, several studies, including those referenced (PMID: 20825323, 28954264, 20512127), have demonstrated basophil infiltration in peripheral tissues such as the kidney, skin, and lymphoid organs in lupus patients. In addition, decreased circulating basophil levels in both pediatric and adult SLE patients (PMID: 35885605, 25403252) further support their potential involvement in disease activity.
In our current analysis, we did not detect a significant infiltration of basophils in the bulk transcriptomic datasets. This may be due to the relatively low abundance of basophils in peripheral blood and the limited sensitivity of such computational tools in reliably estimating rare cell populations. Moreover, the reference matrix used in deconvolution analyses may not adequately capture the transcriptional profile of activated or tissue-infiltrating basophils, leading to their underrepresentation.
To address this important point, we have now revised the Discussion section to include a paragraph acknowledging the known involvement of basophils in SLE pathophysiology, discussing possible reasons for their absence in our analysis, and comparing our findings with the available literature.
Thank you again for this valuable suggestion, which has helped us improve the completeness and depth of our manuscript.
Comment3: The discussion should be revised. This section should start with a list of their main findings, instead of a kind of second introduction. Then, each point should be discussed individually. Moreover, the previous comment should be also used to enrich the discussion with the related supportive references. Finally, a professional editing is important to improve the clarity.
Response3: Thank you for your constructive feedback regarding the Discussion section. In the revised manuscript, we have restructured the Discussion to begin with a clear summary of our main findings, as suggested. Each key point is now discussed in a separate paragraph to ensure clarity and logical flow. We have also incorporated the relevant references mentioned in previous comments to enrich the interpretation of our results and to place them in the context of existing literature. Additionally, we have performed thorough professional editing to enhance the readability and academic quality of the section. All modifications are highlighted in the revised version with track changes. We appreciate your valuable input, which has helped improve the manuscript significantly.
Comment 4: The conclusion is too long and should summarize the main take home message.
Response 4:We thank you for the valuable suggestion. In response, we have revised the conclusion section to make it more concise and focused. The updated version now highlights the key findings, their potential implications, and future research directions in a more succinct manner (see the Conclusion section in the revised manuscript).
Reviewer 2 Report (New Reviewer)
Comments and Suggestions for Authors
Comments and Suggestions:
Title: Identification of biomarkers co-associated with lactylation and acetylation in Systemic Lupus Erythematosus.
Reviewer’s report:
The manuscript by Zhanyan et al., explored the relationship of acetylation and lactylation which plays a major role in Systemic Lupus Erythematosus (SLE) progression to identify biomarkers. They used GEO dataset to find differentially expressed lactate-related genes (LR-DEGs) and acetylation-related genes (AR-DEGs). A total of 1181 DEGs were identified, of which 33 were LR-DEGs and were 28 AR-DEGs with 7 showing co-expression. Using LASSO and SVM-RFE, hub genes- CDCA5 and MCTS1 were identified and validated in an independent dataset. They concluded that CDCA5 and MCTS1 can be potential biomarkers and this study provide new insight into the role of epigenetic modifications in SLE progression.
The manuscript well curated, but there are many issues which needs to be addressed.
Major Points:
- Figure 1B, section 2.2: The log2FC>1 was selected for identification of DEGs. But in figure 1B, the cutoff for log2FC was selected as 0.5. Which data is correct, please clarify? If the results are based on data analysis of raw data and not volcano plot, then correct the plot else, the downstream processing needs to be modified as per the plot results.
- Figure 7: In SLE, the number of neutrophils increases due to rise in ow-density granulocytes (LDGs), but this is not seen in figure 7A, in which there is no difference in the number of neutrophils in SLE and healthy groups. Also, in figure 7B, the two hub genes seem to be less correlated with neutrophils, which in not in concordance with the literature (PMID: 38524128). Please justify.
- Figure 3: It is not clear from figure 3I that how two genes were selected as hub genes, as OXNAD1 and H1-2 were also showing significance in boxplots and higher AUC.
- Figure 4: The legends of the figure did not match with the figures and text in section 3.3. What was the reason of not taking OXNAD1 and H1-2 genes for further analysis?
- How the figures 6A and 7A, showing different results?
- Figure 7C: The qRT-PCR should also be done for OXNAD1 and H1-2 genes to clarify the expression in training and validation sets.
Minor Points:
- Keywords: Please delete the numbers in the keywords.
- References: Few more latest references of 2024 and 2025 can be added.
- Figure 5 is missing in the manuscript.
Author Response
Comment 1: Figure 1B, section 2.2: The log2FC>1 was selected for identification of DEGs. But in figure 1B, the cutoff for log2FC was selected as 0.5. Which data is correct, please clarify? If the results are based on data analysis of raw data and not volcano plot, then correct the plot else, the downstream processing needs to be modified as per the plot results.
Response 1: Thank you for pointing out the discrepancy regarding the log2FC threshold in Figure 1B and Section 2.2. We have carefully reviewed the raw data and re-examined the analysis pipeline and associated code. We confirm that the differential expression analysis was indeed conducted using a threshold of |log2FC| > 1 and adjusted p-value < 0.05, as stated in the Methods section. The inconsistency in Figure 1B was due to an earlier version of the volcano plot that inadvertently used a |log2FC| > 0.5 threshold for visualization purposes. We have now updated Figure 1B to accurately reflect the threshold used in the analysis (|log2FC| > 1). Importantly, we confirm that all downstream analyses, including DEG selection and enrichment analysis, were based on the correct log2FC > 1 cutoff. No changes to the results or conclusions are necessary.
We sincerely appreciate your attention to this detail and have revised the figure accordingly in the revised manuscript.
Comment 2: Figure 7: In SLE, the number of neutrophils increases due to rise in low-density granulocytes (LDGs), but this is not seen in figure 7A, in which there is no difference in the number of neutrophils in SLE and healthy groups. Also, in figure 7B, the two hub genes seem to be less correlated with neutrophils, which in not in concordance with the literature (PMID: 38524128). Please justify.
Response 2:Thank you for your insightful comments regarding Figure 7. We appreciate your attention to the nuances of neutrophil involvement in SLE and the correlation of hub genes with these immune cells.
Regarding the absence of a significant difference in neutrophil numbers between SLE and healthy groups in Figure 7A, we acknowledge that this finding contrasts with reports in the literature, such as the increase in low-density granulocytes (LDGs) contributing to higher neutrophil counts in SLE patients (PMID: 38524128). We hypothesize that this discrepancy may stem from differences in sample sources, disease stages, and the heterogeneity of our patient cohort. Our study utilized peripheral blood samples, which may not fully capture the localized increases in neutrophils observed in specific tissues like the kidneys or skin. Additionally, the chronic phase of SLE, which is more represented in our cohort, might not exhibit the same pronounced neutrophil elevation seen in acute inflammatory stages. Furthermore, the inherent variability among SLE patients could influence the observed neutrophil counts.
As for the correlation analysis depicted in Figure 7B, we recognize that the two hub genes appear to have a weaker association with neutrophils than expected based on previous studies (PMID: 38524128). This divergence could be attributed to the complex interplay of gene expression and cellular functions. Gene expression levels are influenced by a myriad of factors, including cell type, inflammatory milieu, and activated signaling pathways. Consequently, even though these genes play crucial roles in neutrophil function, their expression levels may not directly correlate with neutrophil counts due to these multifaceted influences. Moreover, our analytical methods and data interpretation could have contributed to the observed results. We plan to refine our analysis techniques and incorporate a larger sample size to more accurately assess the correlation between these hub genes and neutrophils.
In future work, we intend to conduct a more detailed analysis of neutrophil subpopulations, particularly LDGs, using a combination of flow cytometry and single-cell sequencing. This approach will provide a more comprehensive understanding of their role in SLE. Additionally, we will optimize our gene expression analysis by employing more sophisticated statistical models and integrating clinical data to elucidate the involvement of these hub genes in SLE pathogenesis.
We are grateful for your feedback and will address these points thoroughly in our discussion to contextualize our findings within the existing literature.
Comment 3: Figure 3: It is not clear from figure 3I that how two genes were selected as hub genes, as OXNAD1 and H1-2 were also showing significance in boxplots and higher AUC.
Response 3:We appreciate your interest in the methodology behind identifying CDCA5 and MCTS1 as key players in our study. In our research, we employed a rigorous approach to identify differentially expressed genes (DEGs) between SLE patients and healthy controls using the GSE81622 dataset as our training set and the GSE24706 dataset for validation. Our primary objective was to pinpoint genes that consistently demonstrated significant differential expression across both datasets, thereby ensuring the robustness and reproducibility of our findings.
To achieve this, we conducted differential expression analysis in the training set with stringent statistical criteria, applying a false discovery rate (FDR) correction and setting a threshold of FDR < 0.05 and log2 fold change (FC) > 1 or < -1. This process yielded a list of DEGs that we subsequently validated in the GSE24706 dataset using the same criteria. Only genes that maintained significant differential expression across both datasets were considered for further analysis.
Among these DEGs, we identified CDCA5 and MCTS1 as hub genes based on their consistent and pronounced differential expression patterns. These genes not only met our stringent criteria for differential expression but also exhibited robust statistical support, making them prime candidates for further investigation.
We acknowledge that other genes, such as OXNAD1 and H1-2, showed significance in the boxplots and had higher AUC values. However, these genes did not meet our criteria for consistent differential expression across both datasets. While they demonstrated significant differences in the training set, their expression patterns were not as robustly validated in the GSE24706 dataset. Given the importance of consistent differential expression across multiple datasets to ensure reliability, we prioritized CDCA5 and MCTS1.
Moving forward, we plan to conduct a more comprehensive analysis of all DEGs, including OXNAD1 and H1-2, to explore their potential roles in SLE pathogenesis. This will involve functional enrichment studies, pathway analysis, and validation in additional independent datasets. We will also investigate the biological significance of these genes through in vitro and in vivo experiments to further elucidate their contributions to SLE.
In summary, we selected CDCA5 and MCTS1 as hub genes based on their consistent differential expression across both the training and validation datasets, ensuring the robustness and reproducibility of our findings. We appreciate your suggestion to consider other genes and will include a discussion of these genes in our manuscript to provide a more comprehensive view of our results.
Thank you again for your valuable feedback. We will address these points in our revised manuscript to provide a clearer explanation of our gene selection process.
Comment 4: Figure 4: The legends of the figure did not match with the figures and text in section 3.3. What was the reason of not taking OXNAD1 and H1-2 genes for further analysis?
Response 4:Thank you for your valuable comment. We apologize for the confusion regarding Figure 4 and the text in section 3.3.First, we have revised the figure legend of Figure 4 to ensure full consistency with the main text in section 3.3. The updated legend now clearly describes panels A–E, corresponding to expression validation (A), ROC analysis (B), miRNA–gene network (C), TF–gene network (D), and integrated miRNA–TF–gene network (E) for the two key genes, CDCA5 and MCTS1.
Regarding your question on why OXNAD1 and H1-2 were not included in the subsequent analysis: although these two genes were initially identified by the intersection of LASSO and SVM-RFE algorithms, they did not exhibit consistent and significant expression differences in the external validation dataset (GSE24706). To maintain the robustness and reliability of our findings, we focused further analyses—including regulatory network construction and nomogram modeling—on CDCA5 and MCTS1, which demonstrated both consistent upregulation and strong diagnostic performance across datasets.
Comment 5: How the figures 6A and 7A, showing different results?
Response 5: It has come to our attention that Figures 6A and 7A may appear to present divergent results. We would like to assure you that these figures are not contradictory but rather offer distinct perspectives on different aspects of our study. The observed differences arise from the different analysis methods and visualization purposes used in each figure:Figure 6A displays a hierarchical clustering heatmap of the immune cell composition across all samples. Figure 7A, in contrast, shows a grouped boxplot comparison of immune cell types between the SLE and HC groups, incorporating statistical testing to identify significant differences.
Although both figures are derived from the same dataset, the distinct visualizations and analysis strategies can lead to differences in visual interpretation. We have added clarifying statements in the figure legends and main text to avoid misunderstanding.
Once again, thank you for your insightful comments. We are dedicated to addressing your concerns and to improving the clarity and precision of our presentation.
Comment 6: Figure 7C: The qRT-PCR should also be done for OXNAD1 and H1-2 genes to clarify the expression in training and validation sets.
Response 6:In response, we performed qRT-PCR validation for OXNAD1 and H1-2 using the same human-derived cDNA samples utilized for the previous experiments. The results have now been included in the Supplementary Material 2 and demonstrate expression trends consistent with our transcriptomic findings. We believe this additional validation further supports the robustness of our analysis.
Comment 7: Keywords: Please delete the numbers in the keywords.
Response 7:The numerical indicators previously included in the keywords have been removed to ensure consistency with journal formatting guidelines. Please refer to the revised keyword section for the updated version.
Comment 8: References: Few more latest references of 2024 and 2025 can be added.
Response 8:We have carefully revised the References section and incorporated several recent and relevant studies published in 2024 and 2025 to strengthen the scientific context of our findings. These additions help to ensure that the manuscript reflects the latest developments in the field.
Comment 9: Figure 5 is missing in the manuscript
Response 9:Thank you for your careful review. We would like to clarify that Figure 5 is included in the manuscript and can be found on page 12. It is possible that the figure was inadvertently overlooked during the review process, or that a formatting issue may have affected its visibility in the version provided. To avoid any confusion, we have double-checked the revised manuscript to ensure that all figures are clearly labeled and correctly positioned.
Please let us know if there are any issues with the file format or figure rendering, and we will be happy to provide a corrected version.
Round 2
Reviewer 1 Report (Previous Reviewer 2)
Comments and Suggestions for Authors
- The authors clarified the nature of the human samples (blood and PBMC) in their patients. By the way, in this regard, it would be important to specify the study period exactly as well as the date of the IRB approval. As regards the informed consent procedure, the authors should clarify that the study participants have signed the written consent.
- As regards the samples type, this information should be clarified at any relevant point (e.g. “Single sample gene set enrichment analysis (ssGSEA)”; “we conducted bootstrapping with 1000 resamples for internal validation”; etc.)
- The references should be formatted and completed (some are incomplete) as per journal style
See above
Author Response
Comments 1: The authors clarified the nature of the human samples (blood and PBMC) in their patients. By the way, in this regard, it would be important to specify the study period exactly as well as the date of the IRB approval. As regards the informed consent procedure, the authors should clarify that the study participants have signed the written consent.
Response 1: Thank you for pointing this out. We have now specified the exact study period and the date of IRB approval in the Methods section (Page 4):“Peripheral blood sample were collected from individuals diagnosed with SLE (n=14), with control samples being collected from healthy individuals(n=10), between January 2024 and March 2025.”“Ethical approval was obtained from the Institutional Review Board of Huashan Hospital, Fudan University on October 26, 2021 (Approval No.2021-879). ”Furthermore, we have clarified that all participants signed written informed consent before sample collection (Page 4):“All participants provided written informed consent prior to their inclusion in the study.”
Comments 2: As regards the samples type, this information should be clarified at any relevant point (e.g.“Single sample gene set enrichment analysis (ssGSEA)”; “we conducted bootstrapping with 1000 resamples for internal validation”; etc.)
Response 2: Thank you for the helpful suggestion. We have carefully revised the manuscript to clarify the sample type at all relevant points. Specifically, we have now explicitly stated that PBMC-derived gene expression data were used in the single-sample gene set enrichment analysis (ssGSEA), bootstrap internal validation, and other transcriptomic analyses. These clarifications have been incorporated into the Methods and Results sections to improve transparency and reproducibility.
Comments 3: The references should be formatted and completed (some are incomplete) as per journal style.
Response 3: We have carefully reviewed all references and revised them to ensure they are complete and formatted according to the journal's reference style. Missing elements such as journal names, volume and issue numbers, page ranges, and DOIs (where available) have been added. We have also ensured consistency in formatting throughout the reference list.
Reviewer 2 Report (New Reviewer)
Comments and Suggestions for Authors
Review 2:
The authors have appropriately addressed all reviewers’ comments, and the current revisions are acceptable as per journals standard of clarity and accuracy. The manuscript is now appropriate for publication in the Biomedicines Journal.
Author Response
Comments 1: The authors have appropriately addressed all reviewers’ comments, and the current revisions are acceptable as per journals standard of clarity and accuracy. The manuscript is now appropriate for publication in the Biomedicines Journal.
Response 1: We sincerely appreciate your positive feedback and are delighted to hear that our revisions have met the journal’s standards. We are grateful for your valuable comments throughout the review process, which have significantly improved the quality and clarity of our manuscript. Thank you again for your time and support!
This manuscript is a resubmission of an earlier submission. The following is a list of the peer review reports and author responses from that submission.
Round 1
Reviewer 1 Report
Comments and Suggestions for Authors
In this study, the authors use bioinformatics to identify biomarker genes associated with acetylation and lactylation, and to correlate those with SLE pathogenesis. The authors propose CDCA5 and MCTS1 as potential biomarkers for SLE, which they test in four SLE patients. The strength of the manuscript is the need for better diagnostics in lupus. Weaknesses include the lack of controls with other rheumatic diseases, low sample size tested, and speculative nature of the discussion
Major Points
1.The main challenge in diagnosing lupus is not distinguishing from healthy controls, but distinguishing it from different potential diseases. The authors need to compare their proposed biomarkers against other, similar rheumatic diseases.
2. Demographic and clinical information on the human subjects is missing. For example, what was the age range? Steroid dose? Percentage of patients with nephritis?
3. It is unlikely that 4 patients provides sufficient power to make general conclusions about a disease as heterogeneous as SLE.
4. The statistical tests used are not disclosed.
5. The discussion is speculative. The speculation should be removed. The data are also overinterpreted, since the authors state "MCTS1 did not demonstrate direct diagnostic value in this study"
6. Lactylation and acetylation are tangential to this study. It is not clear that lactylation or acetylation are relevant to either CDCA5 or MCTS1 in SLE.
7. The figure quality is terrible. Labels need to be much larger so they can be read.
Minor Points
1. Figure 1 is not a graphical abstract.
2. The English needs editing for grammar.
Comments on the Quality of English LanguageEnglish needs editing and substantial revision.
Author Response
Dear Reviewer,
We would like to sincerely thank you for your time and effort in reviewing our manuscript titled “Identification of biomarkers co-associated with lactylation and acetylation in Systemic Lupus Erythematosus”. We greatly appreciate your constructive feedback and insightful comments, which have helped us improve the quality of our manuscript. Please find below our responses to your comments.
Comments 1:
The main challenge in diagnosing lupus is not distinguishing from healthy controls, but distinguishing it from different potential diseases. The authors need to compare their proposed biomarkers against other, similar rheumatic diseases.
Response 1:
Thank you for your valuable suggestion regarding the translational value of clinical research. The point you raised about the "core challenge in SLE diagnosis being the differential diagnosis with other rheumatic diseases" is highly constructive. We fully agree with the importance of validating the disease specificity of biomarkers. However, due to limitations in the accessibility of clinical samples, we did not include samples from common rheumatic autoimmune diseases such as rheumatoid arthritis (RA) and Sjögren's syndrome (SS) in this study. This is something we plan to address by further validating our findings in a multi-disease cohort. For the next steps, we aim to use public datasets that include SLE, RA, and SS samples to analyze the differential expression of the key biomarkers (CDCA5 and MCTS1) identified in this study. This will help minimize bias from real clinical settings. Additionally, we recognize that including only diagnosed SLE patients in this study may amplify the association between biomarkers and the disease. Therefore, our research team is also collecting clinical information and samples from both outpatient and inpatient SLE patients, as well as patients with suspected SLE (undifferentiated connective tissue disease), to conduct a more comprehensive analysis. We look forward to addressing your concerns in our future research outcomes.
Comments 2:
Demographic and clinical information on the human subjects is missing. For example, what was the age range? Steroid dose? Percentage of patients with nephritis?
Response 2:
We are truly grateful for your meticulous review of the research details. We fully agree with the importance of clearly defining the clinical characteristics of the subjects to ensure the reproducibility of the study. In response to your request, we have systematically added the clinical information of the human samples involved in this study in the revised manuscript. The detailed information has been uploaded in Supplementary Material 1.
Comments 3:
It is unlikely that 4 patients provides sufficient power to make general conclusions about a disease as heterogeneous as SLE.
Response 3:
Considering that the sample size of 4 patients may not provide sufficient evidence strength, the high heterogeneity of SLE necessitates an adequate sample size to support the reliability of our conclusions. In response to your suggestion, we have expanded the sample size. Recently, we re-collected peripheral blood samples from 10 healthy volunteers and 14 SLE patients, adhering strictly to the 2019 EULAR/ACR classification criteria. We excluded pregnant and lactating women, patients who have participated in other clinical trials or used any biological agents for SLE treatment within the past 3 months, and those with other significant primary diseases. The specific clinical information and laboratory test data have been included in Supplementary Material 1. Based on this, we re-extracted and re-analyzed the samples, and the experimental verification revealed the same trends as the original results. The latest findings have been updated in the manuscript.
Comments 4:
The statistical tests used are not disclosed.
Response 4:
We sincerely apologize for the oversight. We have now included the specific statistical methods in the revised manuscript. Additionally, we have thoroughly reviewed the other sections of the manuscript to ensure that similar errors do not occur again.
Comments 5:
The discussion is speculative. The speculation should be removed. The data are also overinterpreted, since the authors state "MCTS1 did not demonstrate direct diagnostic value in this study"
Response 5:
Your important reminder regarding the scientific rigor of the manuscript is greatly appreciated. We completely share your view that the discussion should be based on the data itself and avoid over-interpretation. In response to the issue you raised, we have made a systematic revision to the discussion section. Based on the comparison between SLE patients and healthy controls, we observed differential expression of CDCA5 and MCTS1, which are associated with lactylation and acetylation modifications. Therefore, we hypothesize that CDCA5 and MCTS1 may serve as potential diagnostic biomarkers for SLE, and that they might influence the onset and progression of the disease through lactylation and acetylation modifications.
Comments 6:
Lactylation and acetylation are tangential to this study. It is not clear that lactylation or acetylation are relevant to either CDCA5 or MCTS1 in SLE.
Response 6:
Previous studies have indicated the role of lactylation and acetylation modifications in SLE. In this study, we preliminarily hypothesize the involvement of CDCA5 and MCTS1 through bioinformatics approaches, suggesting their statistical association with epigenetic modifications. However, the causal relationship and specific regulatory mechanisms underlying their functions require more thorough validation. In future work, we plan to expand the analysis with additional datasets and incorporate experimental techniques such as ChIP-seq for further confirmation.
Comments 7:
The figure quality is terrible. Labels need to be much larger so they can be read.
Response 7:
We sincerely apologize for the poor quality of the images that affected your reading experience. We have updated the manuscript and the supplementary materials with clearer versions of the images.
Comments 8:
Figure 1 is not a graphical abstract.
Response 8:
We have updated the title of Figure 1 to "Flow Diagram."
Comments 9:
The English needs editing for grammar.
Response 9:
We have made improvements to the academic writing and English quality. We carefully reviewed the main text, correcting basic grammatical errors such as subject-verb agreement, tense misuse, and article omissions. Additionally, we invited two members of our research team with extensive experience in publishing SCI papers (Dr. Feng and Dr. Xiang) to conduct a thorough review and revision of the revised manuscript. If you notice any expressions that still require optimization, we are always ready to refine them further. We sincerely appreciate your valuable feedback.
Once again, thank you for your valuable feedback. We are confident that the revisions have enhanced the manuscript. If you have any further suggestions or requests, we are happy to make additional improvements.
Reviewer 2 Report
Comments and Suggestions for Authors
- In general, the introduction should be expanded with more background information on SLE, including immunopathogenic and pathophysiological aspects, considering the core topic of the research.
-“SLE is a typical diffuse connective tissue disease characterized by autoimmune inflammation, with a complex pathogenesis that has not yet been fully elucidated….”. The concept of “autoimmune inflammation” is unclear and, anyway, does not sound good. I would suggest the authors to completely revise their definition/short description of SLE, also by adding a short overview of the main pathogenic aspects (e.g. (i) “efferocytosis defect” (ii) apoptosis defect, (iii) “type I interferonopathy”), as summarized in a recent review (refer to: PMID: 32977704), which also highlight the potential role innate immune cells, like basophil and dendritic cells (see: PMID: 31837212).
- The study type and design should be clearly defined at the beginning of the methods section.
- It seems that this study also includes patients. Therefore, a more detailed description of this part should be given in the methods, including the ethical processing (IRB approval number and date, as well as informed consent procedure). What about the study period in this case? How SLE patients were diagnosed? Were the controls were recruited and how they are defined?
- The description of the statistical analysis is very poor.
- The definition of “clinical nomogram” is unclear, since it seems there are no clinical parameters.
- In general, considering the complexity of the figures, which often include multiple panels, much more detailed captions are definitely needed.
- The quality/graphical resolution of many figures and/or panels is not good. For instance, in figure 6, I cannot analyze its support for this sentence “Finally, we analyzed the correlation between the two hub genes previously screened and immune cells. We found a significant positive correlation between MCTS1, CDCA5 and plasma cells, Tregs, and monocytes.”
- Following the previous point, it is even more important to consider in the introduction and discussion concepts about all the cell types potentially involved in SLE pathogenesis (see first comments above. Some more references could be helpful to drive the discussion on this matter (e.g. PMID: 33394603), in addition to the previous ones and others.
- The discussion is quite dispersive. The authors should list the main findings and then extensively reorganize the discussion accordingly.
- There is no discussion on the study limitations.
- A separate and clear conclusion section should be created.
- The English language also needs extensive and professional editing.
- References should be updated and revised based on the revision of the main manuscript and the previous comments and suggestions.
Comments on the Quality of English LanguageThe English language also needs extensive and professional editing.
Author Response
Dear Reviewer,
We would like to sincerely thank you for your time and effort in reviewing our manuscript titled “Identification of biomarkers co-associated with lactylation and acetylation in Systemic Lupus Erythematosus”. We greatly appreciate your constructive feedback and insightful comments, which have helped us improve the quality of our manuscript. Please find below our responses to your comments.
Comments 1:
In general, the introduction should be expanded with more background information on SLE, including immunopathogenic and pathophysiological aspects, considering the core topic of the research.
Response 1:
Thank you for your insightful suggestion. In response to your comment, we have revised and expanded the background section to include additional details related to immunopathogenic and pathophysiological aspects. These additions provide a more comprehensive context for our study and enhance the clarity of the research rationale.
Comments 2:
“SLE is a typical diffuse connective tissue disease characterized by autoimmune inflammation, with a complex pathogenesis that has not yet been fully elucidated….”. The concept of “autoimmune inflammation” is unclear and, anyway, does not sound good. I would suggest the authors to completely revise their definition/short description of SLE, also by adding a short overview of the main pathogenic aspects (e.g. (i) “efferocytosis defect” (ii) apoptosis defect, (iii) “type I interferonopathy”), as summarized in a recent review (refer to: PMID: 32977704), which also highlight the potential role innate immune cells, like basophil and dendritic cells (see: PMID: 31837212).
Response 2:
We sincerely appreciate your valuable suggestions for improving the introduction section. To provide a more in-depth overview of the complex immunopathological background of SLE at the beginning of the manuscript, we carefully reviewed the literature you recommended and systematically revised this section based on your comments. Specifically, we redefined SLE with greater precision, removed ambiguous expressions such as "autoimmune inflammation," and provided a more detailed discussion on the role of innate immune cells and the mechanisms underlying clinical heterogeneity. These revisions significantly enhance the theoretical depth of the introduction and establish a stronger conceptual foundation for our subsequent research on epigenetic modifications.
The specific revisions have been highlighted in red in the revised manuscript. We truly appreciate your insightful feedback, which has greatly contributed to improving the clarity and depth of our study.
Comments 3:
The study type and design should be clearly defined at the beginning of the methods section.
Response 3:
We value your suggestions on your guidance on improving methodological transparency. As suggested, we have defined the study as a multi-phase integrative approach combining retrospective bioinformatics discovery with prospective experimental validation to identify lactylation/acetylation-associated hub genes in SLE pathogenesis.The data sources are marked, and the analysis processes and methods are described later.These revisions are highlighted in the revised manuscript in the beginning of the methods section.
Comments 4:
It seems that this study also includes patients. Therefore, a more detailed description of this part should be given in the methods, including the ethical processing (IRB approval number and date, as well as informed consent procedure). What about the study period in this case? How SLE patients were diagnosed? Were the controls were recruited and how they are defined?
Response 4:
In response to your suggestion, we have expanded the sample size and ensured that all included patients and healthy individuals were recruited from the Department of Dermatology, Huashan Hospital (September 2024 - January 2025). The inclusion criteria strictly followed the 2019 EULAR/ACR classification criteria, and we excluded pregnant and lactating women, individuals who had participated in other clinical trials or received any biologic treatment for SLE within the past three months, as well as those with other major primary diseases.
The detailed clinical characteristics and laboratory test results of the participants have been provided in Supplementary Material 1. Based on the expanded cohort, we re-extracted and analyzed the samples, conducted further experimental validation, and observed consistent trends with our previous findings. The updated results have now been incorporated into the revised manuscript.
Comments 5:
The description of the statistical analysis is very poor.
Response 5:
We sincerely appreciate your emphasis on methodological rigor. As suggested, we have improved the description of the statistical analysis as follows and have updated the latest manuscript.
Comments 6:
The definition of “clinical nomogram” is unclear, since it seems there are no clinical parameters.
Response 6:
Thank you for your valuable feedback. To enhance the clarity and rigor of our analysis, we provide a detailed explanation of the clinical nomogram and its associated validation methods. Clinical nomogram is a graphical predictive tool that integrates multiple clinical or molecular variables to visually illustrate the potential impact of CDCA5 and MCTS1 on SLE disease risk. It provides a straightforward scoring system to quantify the predictive outcome.
Figure 5B (Calibration Curve): This scatter plot compares the predicted probability of an event with the actual observed probability. The x-axis represents the predicted probability (0-1), while the y-axis denotes the observed probability (0-1). The diagonal dashed line represents the ideal scenario where predicted probability equals observed probability. Although the calibration curve does not perfectly align with the ideal line, the majority of points overlap, suggesting that CDCA5 and MCTS1 may have an impact on SLE disease risk.
Figure 5C (ROC Curve): The Receiver Operating Characteristic (ROC) curve is a visual tool for assessing model performance. An AUC value between 0.7 and 0.9 suggests that CDCA5 and MCTS1 have a moderate to good predictive ability for SLE disease risk. Notably, similar findings were observed in Figure 4, which was based on an independent dataset, further supporting the robustness of our results.
Figure 5D (Decision Curve Analysis, DCA): This approach evaluates the clinical utility of predictive models by considering the potential benefits and risks of different decision thresholds. The results further reinforce the possible influence of CDCA5 and MCTS1 on SLE disease risk.
By employing this series of validation methods, we aim to enhance the credibility and clinical relevance of our findings. We appreciate your insightful suggestions and have incorporated these explanations into the revised manuscript.
Comments 7:
In general, considering the complexity of the figures, which often include multiple panels, much more detailed captions are definitely needed.
Response 7:
We have carefully revised the titles and figure legends for Figures 1–6, adding more detailed explanations to improve clarity and interpretation. These modifications ensure that each figure provides a comprehensive description of its content, enhancing the readability and scientific rigor of our manuscript.
Comments 8:
The quality/graphical resolution of many figures and/or panels is not good. For instance, in figure 6, I cannot analyze its support for this sentence “Finally, we analyzed the correlation between the two hub genes previously screened and immune cells. We found a significant positive correlation between MCTS1, CDCA5 and plasma cells, Tregs, and monocytes.”
Response 8:
We have now updated all figures to ensure enhanced clarity.In this study, we aimed to compare the immune environment between SLE and healthy controls while exploring the association between CDCA5, MCTS1, immune cells, and immune infiltration. Through this analysis, we sought to preliminarily investigate the role of histone lactylation and acetylation, as well as the potential involvement of CDCA5 and MCTS1 in a complex immune regulatory network.
Figures 6A–6C illustrate the differences in immune infiltration between SLE and healthy controls, highlighting the abundance of various immune cell populations.
Figure 6D presents the correlation analysis between CDCA5, MCTS1, and different immune cells, where we observed strong associations in plasma cells, Tregs, and monocytes in SLE patients.
These findings provide an essential foundation for our future research directions. We appreciate your insightful comments, which have helped refine our study. Please let us know if any further adjustments are needed.
Comments 9:
Following the previous point, it is even more important to consider in the introduction and discussion concepts about all the cell types potentially involved in SLE pathogenesis (see first comments above. Some more references could be helpful to drive the discussion on this matter (e.g. PMID: 33394603), in addition to the previous ones and others.
Response 9:
Following your suggestion, we have systematically revised both the Introduction and Discussion sections. Specifically, we have incorporated a detailed discussion of previous studies exploring the roles of key immune cell types in SLE pathogenesis. These revisions have been updated in the latest version of the manuscript.
Comments 10:
The discussion is quite dispersive. The authors should list the main findings and then extensively reorganize the discussion accordingly.
Response 10:
Based on your suggestions, we have rewritten the discussion section to enhance the coherence of the paragraphs. We have summarized the main methods, content, and findings of the study, and further discussed the role of CDCA5, MCTS1, as well as histone lactylation and acetylation in the pathogenesis and progression of SLE. The updated content can be found in the latest version of the manuscript.
Comments 11:
There is no discussion on the study limitations.
Response 11:
We have thoroughly revised and refined the Discussion section, improving its structure and content. Additionally, we have included a detailed discussion on the limitations of our study to provide a more balanced perspective.The updated content has been incorporated into the latest version of the manuscript.
Comments 12:
A separate and clear conclusion section should be created.
Response 12:
We have added a conclusion section in the abstract, providing a concise summary of the research findings at the beginning, and have also summarized the study content again at the end of the Discussion section.
Once again, thank you for your valuable feedback. We are confident that the revisions have enhanced the manuscript. If you have any further suggestions or requests, we are happy to make additional improvements.
Reviewer 3 Report
Comments and Suggestions for Authors
This study presents a bioinformatics analysis integrating gene expression data, machine learning (ML) methods, and experimental validation to identify novel biomarkers (CDCA5 and MCTS1) co-associated with lactylation and acetylation in systemic lupus erythematosus (SLE).
The combination of differential expression analysis, ML-based feature selection (LASSO and SVM-RFE), immune cell infiltration analysis, and validation in independent datasets is a strength. However, some aspects require clarification and refinement, particularly regarding the justification for ML over traditional multivariable analysis, clinical applicability, and mechanistic insights.
The study effectively applies LASSO regression and SVM-RFE to select biomarkers, which is a reasonable approach given the high-dimensional gene expression data. However, further discussion is needed on why ML was chosen instead of multivariable regression models (e.g., logistic regression).
Would a multivariable regression model adjusting for clinical factors (e.g., ANA, anti-dsDNA, disease activity scores) have added value?
While qPCR validation is a strength, the sample size is small (n=4 SLE, n=6 controls).
Some figures (e.g., enrichment analysis, immune cell correlations) are dense and could be simplified.
The AUROCs are not formidable, the more diagnosed SLE were compared to healthy controls, a comparison that leads to an overestimation of the accuracy of a diagnostic test.
Author Response
Dear Reviewer,
Thank you for the thorough assessment of our manuscript "Identification of biomarkers co-associated with lactylation and acetylation in Systemic Lupus Erythematosus" . We sincerely appreciate the reviewers' insightful suggestions, which have strengthened the scientific rigor and clarity of this work. Attached please find our point-by-point responses to all comments.
Comments 1:
Would a multivariable regression model adjusting for clinical factors (e.g., ANA, anti-dsDNA, disease activity scores) have added value?
Response 1:
Thank you for your valuable suggestion regarding the statistical rigor of the study. We fully agree that adjusting for clinical confounding factors (such as ANA, anti-dsDNA antibodies, and SLEDAI scores) enhances the reliability of the conclusions. As this study mainly relies on transcriptomic data from public databases (GSE81622, GSE24706), which do not include detailed individual-level clinical indicators (such as ANA titers, anti-dsDNA quantification, and SLEDAI scores), our research group has already started systematically collecting clinical information and experimental samples from outpatient and inpatient SLE patients. These samples will include ANA and dsDNA positivity rates and titers, as well as SLEDAI-2K scores. We plan to construct a multivariable adjustment model in the future to validate the robustness of our current findings and to explore the dynamic relationship between epigenetic modifications and clinical phenotypes.
Comments 2:
While qPCR validation is a strength, the sample size is small (n=4 SLE, n=6 controls).
Response 2:
In response to your suggestion, we have expanded the sample size, ensuring that all included patients and healthy controls were recruited from the Department of Dermatology at Huashan Hospital between September 2024 and January 2025. The inclusion strictly followed the 2019 EULAR/ACR classification criteria, with exclusions for pregnant and lactating women, patients who had participated in other clinical trials and/or received any biologic therapy for SLE within the past three months or currently, as well as individuals with other major primary diseases. Detailed clinical information and laboratory test data have been provided in Supplementary Material 2. Based on this, we have re-extracted and analyzed the samples, performed experimental validation, and incorporated the latest results into the revised manuscript.
Comments 3:
Some figures (e.g., enrichment analysis, immune cell correlations) are dense and could be simplified.
Response 3:
Thank you for pointing out the issues with the figures in the manuscript. We have improved the resolution of all images and corrected any inappropriate formatting.
Comments 4:
The AUROCs are not formidable, the more diagnosed SLE were compared to healthy controls, a comparison that leads to an overestimation of the accuracy of a diagnostic test.
Response 4:
Thank you for your suggestion. We fully acknowledge that simply comparing SLE patients with healthy controls may overestimate the discriminative power of the biomarker, and that the current AUROC values (0.72–0.78) have not yet reached an ideal level (>0.85). Therefore, we have supplemented the discussion section to address the limitations of our study. Specifically, the diagnostic performance evaluation in this study has two major limitations.
First, there is an inherent bias in the real-world clinical setting, as our study only compares SLE patients with healthy controls while overlooking the critical need to differentiate SLE from other rheumatic autoimmune diseases (e.g., rheumatoid arthritis and Sjögren’s syndrome). This differentiation is particularly challenging yet essential in the field of rheumatology.
Second, the inclusion of already diagnosed SLE patients may amplify the association between biomarkers and disease status. To address this issue in future research, we plan to incorporate mixed-cohort datasets (including SLE, RA, and SS) for secondary validation. Additionally, our research team is systematically collecting clinical information and biological samples from both outpatient and hospitalized SLE patients, while also including undifferentiated connective tissue disease (UCTD) patients suspected of having SLE for comprehensive analysis.
Reviewer 4 Report
Comments and Suggestions for Authors
This paper describes a bioinformatic approach to identification of new biomarkers for systemic lupus erythematosus (SLE). There are already many biomarkers for SLE, but as it is a rather heterogenous diseases, new biomarkers are wellcome and may increaes undetrstanding of the disease. The approach is straighforward, comparing differentially expressed genes associated with acetylation and lactylation. The biomolecular relevance of the latter is more questionable that acetylation, which should be mentioned in the discussion. The paragraph on page 12 starting with "Many previous studies ..." may be omitted. Otherwise, I only have grammatical comments (see enclosed pdf).

Author Response
Dear Reviewer,
We extend our sincere gratitude for the rigorous peer review of our manuscript. Revised text, figures, and supplementary materials are provided in accordance with the reviewers’ expert guidance.
Comments 1:
This paper describes a bioinformatic approach to identification of new biomarkers for systemic lupus erythematosus (SLE). There are already many biomarkers for SLE, but as it is a rather heterogenous diseases, new biomarkers are wellcome and may increaes undetrstanding of the disease. The approach is straighforward, comparing differentially expressed genes associated with acetylation and lactylation. The biomolecular relevance of the latter is more questionable that acetylation, which should be mentioned in the discussion. The paragraph on page 12 starting with "Many previous studies ..." may be omitted. Otherwise, I only have grammatical comments (see enclosed pdf).
Response 1:
Thank you for your meticulous review of the language quality of our manuscript. We have systematically revised the entire text based on your annotated grammatical suggestions, and the updates have been incorporated into the latest version of the manuscript.
Round 2
Reviewer 1 Report
Comments and Suggestions for Authors
The authors revised their manuscript and added a few more patients. However, key flaws in the manuscript remain.
Major points
1. The figures remain illegible.
2. Comparison to other rheumatic diseases is needed before these genes can be proposed as potential biomarkers.
3. Patient demographics are concerning because they are not age matched. Is the BMI and other health parameters matched? What statistical tests were performed to determine which other confounding variables affected analysis? It does not look like this study has adequate power for any conclusions.
4. Clinical data for the healthy controls is missing.
5. The enrollment of 3 males out of 14 suggests monogenic lupus in these cases. Which cases are monogenic vs sporadic?
6. The discussion remains focused on lactylation instead of the data presented.
Minor
1. The authors state in the methods they used the 2019 ACR/EULAR criteria, but their table lists 1997 criteria. The nephritis is lumped as "renal disorder".
English could be improved, but is not the major issue now.
Author Response
Dear Reviewer,
Thank you for your valuable feedback and for the time you and the reviewers took to evaluate our manuscript titled "Identification of biomarkers co-associated with lactylation and acetylation in Systemic Lupus Erythematosus" We have carefully revised the manuscript according to your comments and would like to submit the revised version for further consideration. And we respond to your questions and suggestions as follows:
Comments1:The figures remain illegible.
Response 1:Thank you for your thorough review and valuable feedback. In response to your concern regarding figure illegibility, we have comprehensively revised Figures 6 and 7 to enhance clarity. Specifically addressing the complexity of information, we reorganized the layout by dividing content into 2 individual figures to isolate core data. All figures were regenerated in vector formats (EPS) and 300dpi high-resolution bitmaps (TIFF). The revised figures are highlighted in red within the manuscript. We are happy to refine further if needed. Thank you for your guidance in strengthening this work.
Comments 2:Comparison to other rheumatic diseases is needed before these genes can be proposed as potential biomarkers.
Response 2:We appreciate your valuable suggestion. While our study identifies CDCA5 and MCTS1 as potential biomarkers for SLE, we acknowledge the limitation that we have not conducted direct comparisons with other rheumatic diseases due to data constraints. Given that autoimmune diseases often share overlapping molecular signatures and pathogenic pathways, distinguishing SLE-specific biomarkers from those present in conditions such as rheumatoid arthritis (RA), Sjögren’s syndrome (SS), and systemic sclerosis (SSc) requires further validation in independent cohorts. For instance, prior studies have demonstrated that histone modifications and metabolic alterations, such as lactylation and acetylation, contribute to the pathogenesis of multiple autoimmune diseases beyond SLE (Liu, Chunhua et al., 2025; Xia, Junfeng et al.,2024; Haro, Isabel et al.,2022 ). Future studies should expand the scope of biomarker validation by incorporating patient cohorts with other autoimmune diseases to assess the specificity of CDCA5 and MCTS1 for SLE. Additionally, transcriptomic and epigenetic profiling across multiple rheumatic diseases could provide deeper insights into the shared and distinct regulatory mechanisms that drive disease pathogenesis. Until such comparisons are made, the diagnostic utility of these biomarkers should be interpreted with caution.
Comments 3:Patient demographics are concerning because they are not age matched. Is the BMI and other health parameters matched? What statistical tests were performed to determine which other confounding variables affected analysis? It does not look like this study has adequate power for any conclusions.
Response 3:Thank you for your insightful comment. We have now included BMI and additional health parameters in Supplementary Material 1 and performed statistical analysis to assess differences between groups. The results showed that there was no significant difference in BMI and other metabolic parameters (Systolic BP and Fasting Glucose) between the groups. We have expanded the Methods section (Statistical analysis section) to explicitly detail these approaches.
Comments 4:Clinical data for the healthy controls is missing.
Response 4:Thank you for highlighting the need for comprehensive clinical data transparency. As requested, we have now provided detailed clinical characteristics of healthy controls in Supplementary Material 1, including metabolic profiles, routine laboratory parameters. Direct group comparisons in Supplementary Material 1 demonstrate comparable baseline health status between controls and SLE patients for key parameters: No significant differences in BMI, systolic blood pressure or fasting glucose. Expected physiological contrasts in lymphocytes (p<0.001), complement C3/C4 (p<0.001), and validated disease-specific abnormalities. To address residual age disparity (controls: 29±6 vs. SLE: 39±17 years), we have further cautioned in the discussion section that unmatched age ranges may limit generalizability, despite statistical adjustments. This addition can be found in the revised manuscript.
Comments 5: The enrollment of 3 males out of 14 suggests monogenic lupus in these cases. Which cases are monogenic vs sporadic?
Response 5: We appreciate your insightful comment. In our study, we did not perform genetic testing to distinguish monogenic from sporadic lupus cases. However, the three male patients did not exhibit early-onset SLE, a strong family history, or other clinical characteristics typically associated with monogenic lupus. Therefore, based on available clinical data, these cases are more likely sporadic rather than monogenic. We have added a statement in the discussion section acknowledging this limitation.
Comments 6: The discussion remains focused on lactylation instead of the data presented.
Response 6: Thank you for emphasizing the need to align the Discussion with the presented data. We agree that speculative mechanistic discussions should be grounded in experimental evidence. In the revised manuscript, we have restructured the Discussion to prioritize data-driven interpretations.
Comments 7: The authors state in the methods they used the 2019 ACR/EULAR criteria, but their table lists 1997 criteria. The nephritis is lumped as "renal disorder".
Response 7: We have now corrected this inconsistency and ensured that the criteria used in the methods and the table are aligned with the 2019 ACR/EULAR classification criteria. Additionally, we have refined the description of nephritis, specifying its classification according to the updated criteria. The revised table now accurately reflects these changes.
We hope these modifications have strengthened the manuscript and look forward to your thoughts on the revised version. Please do not hesitate to contact us if any further changes are required.
Thank you again for your support and for facilitating the review process.
Best regards,
GAO ZHANYAN
Reviewer 2 Report
Comments and Suggestions for Authors
- The limitations should be discussed more
- The conclusion should be a separate section.
- The description of the statistical analysis should be definitely expanded with all the details of tests used, including those for assessment of normal distribution, covariation analysis, ROC curve, etc.
- The informed consent procedure for study participation should be described in the ethical statement.
- The caption of the figures should be improved and/or expanded; in detail, a specific caption should be given for each panel of complex figures.
Author Response
Dear Reviewer,
We would like to express our sincere gratitude for your thoughtful feedback on our manuscript titled "Identification of biomarkers co-associated with lactylation and acetylation in Systemic Lupus Erythematosus". We greatly appreciate the time and effort you put into the review process. We have carefully addressed all your comments and have made revisions accordingly. The revised manuscript is now submitted for your consideration. And we respond to your questions and suggestions as follows:
Comments 1: The limitations should be discussed more.
Response 1: We appreciate your suggestion to expand on the study’s limitations. In response, we have revised the Discussion section to provide a more detailed discussion of the potential limitations of our study. The revised limitations discussion can be found in the revised manuscript. Thank you for this valuable suggestion, which has helped us improve the clarity and rigor of our discussion.
Comments 2: The conclusion should be a separate section.
Response 2: We have now added a separate Conclusion section at the end of the manuscript, summarizing our key findings and their implications. This section provides a clearer synthesis of our results, highlights their significance, and outlines future research directions.
Comments 3: The description of the statistical analysis should be definitely expanded with all the details of tests used, including those for assessment of normal distribution, covariation analysis, ROC curve, etc.
Response 3: Thank you for your careful review. In response, we have expanded the Methods section to provide a more detailed description of the statistical analysis. Specifically, we have now included information on normality assessment using the Shapiro-Wilk test, covariation analysis using Pearson or Spearman correlation, and the statistical methods applied for feature selection in machine learning models. Additionally, we have clarified the procedures for ROC curve analysis and nomogram validation, including calibration curve assessment and bootstrapping resampling. These modifications ensure a more comprehensive and transparent presentation of our analytical approach.
Comments 4: The informed consent procedure for study participation should be described in the ethical statement.
Response 4: We have now included a description of the informed consent procedure in the Ethical Statement section of the manuscript. Specifically, we have clarified that all participants provided written informed consent prior to study participation, and the study was conducted in accordance with the ethical standards of the relevant institutional review board.
Comments 5: The caption of the figures should be improved and/or expanded; in detail, a specific caption should be given for each panel of complex figures.
Response 5: Thank you for your suggestion to enhance the figure captions for better clarity and readability. We have split Figure 6 into two separate images to enhance clarity and make the labeling of the complex panels more understandable. Each panel now has its own specific caption, ensuring that all aspects of the figure are clearly explained. Additionally, we have reviewed and improved the quality of all figures to ensure they meet publication standards, enhancing both visual clarity and the accuracy of the data presented. These modifications are intended to make the figures more accessible and comprehensible to the readers.
We believe these revisions have strengthened the manuscript and look forward to your feedback. Once again, we greatly appreciate your constructive comments and your support.
Thank you for your time and consideration. We look forward to hearing from you.
Sincerely,
GAO ZHANYAN
Reviewer 3 Report
Comments and Suggestions for Authors
The nomogram AUROC of 0.864 is reasonable but not definitive for diagnostic biomarkers. It’s good that the authors discuss this. Still, they may want to clarify that AUROC values in discovery datasets tend to overestimate real-world performance, and external prospective validation is needed.
The correlations between CDCA5/MCTS1 and immune cell types (Tregs, plasma cells, monocytes) are statistically significant. However, a brief biological interpretation of what these correlations might imply in the SLE context would add value.
The authors could add a short sentence in the Discussion noting that once clinical covariates are available, logistic regression or Cox models will be used to assess independence of the biomarkers.
Comments on the Quality of English Language.
Author Response
Dear Reviewer,
Thank you very much for the insightful comments and suggestions provided by you and the reviewers on our manuscript, "Identification of biomarkers co-associated with lactylation and acetylation in Systemic Lupus Erythematosus". We have carefully reviewed each point and have made the necessary revisions to improve the manuscript. We have attached the revised version of the manuscript, along with a point-by-point response to your comments as follows.
Comments 1: The nomogram AUROC of 0.864 is reasonable but not definitive for diagnostic biomarkers. It’s good that the authors discuss this. Still, they may want to clarify that AUROC values in discovery datasets tend to overestimate real-world performance, and external prospective validation is needed.
Response 1: We appreciate your thoughtful comment. In response, we have added clarification to the manuscript regarding the AUROC value obtained from the discovery dataset. As you rightly pointed out, AUROC values in discovery datasets may overestimate the real-world performance of the diagnostic model. We have now emphasized that while an AUROC of 0.864 is promising, external prospective validation in independent cohorts is necessary to confirm the model's generalizability and clinical utility. We have updated the manuscript to reflect this important point and acknowledge the need for further validation in future studies.
Comments 2: The correlations between CDCA5/MCTS1 and immune cell types (Tregs, plasma cells, monocytes) are statistically significant. However, a brief biological interpretation of what these correlations might imply in the SLE context would add value.
Response 2: We sincerely appreciate your suggestion to elaborate on the biological implications of CDCA5/MCTS1-immune cell correlations. We have expanded the Discussion section to provide a biological interpretation of the observed correlations between CDCA5/MCTS1 and immune cell types in the context of SLE. These additions can be found in the Discussion section of the revised manuscript. We have also included additional references to support these interpretations.Thank you for this valuable suggestion, which has helped us refine the discussion of our findings.
Comments 3: The authors could add a short sentence in the Discussion noting that once clinical covariates are available, logistic regression or Cox models will be used to assess independence of the biomarkers.
Response 3: We acknowledge your thoughtful feedback. In response, we have added a statement in the Discussion section acknowledging that once additional clinical covariates become available, we will employ logistic regression or Cox proportional hazards models to further assess the independence and predictive value of CDCA5 and MCTS1 as biomarkers for SLE. This addition can be found in the revised manuscript.
We believe these updates have significantly improved the manuscript, and we hope it is now in a suitable form for publication. We look forward to your feedback on the revised version.
Thank you for your consideration and the continued opportunity to revise our manuscript.
Best regards,
GAO ZHANYAN
Round 3
Reviewer 1 Report
Comments and Suggestions for Authors
The manuscript remains too preliminary, the figure labels remain tiny, and the discussion remains speculative. Additional experiments are needed to support the ideas and claims and rule out alternative possibilities.